# The Effect of Plant Growth Promoting Rhizobacteria (PGPRs) on Yield and Some Quality Parameters during Shelf Life in White Button Mushroom (*Agaricus bisporus* L.)

**DOI:** 10.3390/jof8101016

**Published:** 2022-09-27

**Authors:** Erkan Eren

**Affiliations:** Bergama Vocational Training School, Mushroom Program, Ege University, Bergama 35700, İzmir, Turkey; erkan.eren@ege.edu.tr

**Keywords:** beneficial bacteria, yield, total phenolics, antioxidant capacity, postharvest quality

## Abstract

The use of different bacteria that increase yield and quality in plant production has become common since the 1990s. However, effects of plant growth promoting rhizobacterial (PGPR) treatments during the cultivation period of white button mushroom on quality during marketing duration are not known exactly. This study was carried out to determine the effects of different PGPRs in compost medium on mushroom yield and quality. For this reason, *Azospillum lipoferum*, *Bacillus megaterium*, *Frateuria aurantia* and *Thiobacillus thiooxidans*, for promoting nitrogen, phosphorus, potassium and sulfur transport, respectively, were applied at a 3 mL per m^2^ concentration on the 12th day of the spawn run period. Control groups were treated with only water. *Azospillum lipoferum* increased yield at a rate of 33.3% by enhancing mushroom number per unit area. Shelf life characteristics were observed in mushrooms after storage at 1 °C and at 20 ± 1 °C for 2 days. *Bacillus megaterium*, *Frateuria aurantia* and *Thiobacillus thiooxidans* treatments decreased weight loss and loss in cap firmness, total phenolic compounds and antioxidant capacity during shelf life. However, *Azospillum lipoferum* for increased yield and *Frateuria aurantia* and *Bacillus megaterium* for maintaining postharvest quality were promising treatments during shelf life.

## 1. Introduction

Increasing public awareness of human nutrition has led to an increased interest in mushrooms, which are good protein, mineral and vitamin sources. Mushrooms are preferred by humans not only for their bioactive compounds such as vitamins, minerals, polyphenolics, flavonoids and organic acids but also their medicinal characteristics [1]. The amount of cultivated mushroom consumption has increased gradually in the last 15 years, and per capita consumption has increased from 1 kg to 4 kg [2]. *Agaricus* species has the highest production and consumption in the world, and within this species, *Agaricus bisporus* L., also known as the white button mushroom, is the first [3].

The physiological effects of beneficial bacteria that promote plant growth have attracted attention since the beginning of the 20th century [4,5,6]. Beneficial bacteria are effective in growth hormone and organic compound production, germination, root development, nutrient uptake, productivity and resistance to stress and disease [7,8]. The use of bacteria called biofertilizers or control agents in agricultural production increased after the 1990s. Recently, the scope of biological fertilization has expanded and rhizobacteria, which are free-living and encourage plant growth, have been used as biological warfare agents or biofertilizers. Bacteria increase the production of growth hormones in plants and increase the uptake of some minerals, inhibit ethylene synthesis and stimulate resistance against diseases by contributing to production of siderophores, vitamins and antibiotics [9,10].

In order to develop resistance in plants, biotic stimulants (bacteria, fungi, viruses, nematodes, etc.) or abiotic stimulants (salicylic acid, glycine, methyl jasmonate, ethylene, some herbicides, etc.) have been used in many cultivated plants against many pathogens [11]. Biological control has been carried out with plant growth promoting rhizobacteria (PGPRs) thanks to different mechanisms, such as competition, promotion of antibiosis and resistance [12]. Some PGPRs produce siderophores that prevent iron uptake for harmful microorganisms, and therefore, besides reducing the pathogenic effect of these harmful microorganisms, they also suppress microorganisms that cause disease by producing substances that are harmful to pathogens such as antibiotics and hydrogen cyanide (HCN) [13].

Today, the importance of microorganisms that spontaneously form in the root zone and interact beneficially with plant root parts is increasing. Root bacteria offer an increase in plant growth and yield, as well as their antagonistic effects [14]. Soil bacteria can act indirectly by preventing some harmful effects of pathogenic microorganisms in plant development or directly by synthesizing a compound produced by the microorganism and facilitating the uptake of nutrients in the root region [15]. Although the mechanisms of root bacteria stimulating plant growth are not fully explained, they increase nutrient uptake by providing resistance to biotic and abiotic stress conditions and positively affecting the root activity of the plant [16].

Bacteria that promote plant growth have functions such as seed germination, root development, the plant’s use of water, production of growth hormones, changing the microbial balance in the root zone in favor of beneficial microorganisms, indirectly affecting plant growth by regulating the mineral substance ratio, preventing bacterial and fungal diseases and protecting against viral diseases [17]. These beneficial bacteria have been widely used in horticultural production, especially for higher yield and quality and disease control in recent years [18,19,20,21,22]. Although there are many studies on the effect of beneficial bacteria on fruit and vegetable production around the world, studies on their use in mushroom production are very limited. In these studies, PGPR applications stimulated mushroom sporophore formation.

Today, practices that are called environmentally friendly are widely used in agriculture. Microorganisms beneficial for human and environmental health, one of the most important of these applications, are seen to be widely used [23]. Soil bacteria with different mechanisms that develop in the root zone affect plant growth positively. Considering the working mechanism of a similar effect, it is thought that inoculating compost will give positive results on mushroom yield and quality by encouraging the growth of mycelia in the compost as well as promoting underground root growth. For example, inoculation of *Pseudomonas putida* solution increased the yield of white button mushroom at a rate of 12.1% [24].

In the current study, the effects of different root bacterial solutions inoculated into compost on the quality of white button mushrooms during shelf life periods in addition to yield at harvest and cold storage were studied.

## 2. Materials and Methods

### 2.1. Production of Mushrooms and PGPR Applications

The study was carried out in the production room of the Research and Application unit at Ege University Bergama Vocational School Mushroom Program. The compost, which is the culture medium used in the experiment, was obtained from the PEMA MANTAR Agriculture enterprise located in the old Foça district of İzmir, inoculated with commercial mycelium belonging to the company Sypra. *Azospillum lipoferum* (AL), *Bacillus megaterium* (BM), *Frateuria aurantia* (FA) and *Thiobacillus thiooxidans* (TT), for promoting nitrogen, phosphorus, potassium and sulfur transport in the growing mushrooms, respectively, were inoculated into the compost on the 12th day of the spawn run period with 0.5 L of water, with 3 mL per m^2^ in each plot. As a control medium, only 0.5 mL of water was applied to the compost without inoculating bacteria.

The composts made by inoculating mycelium spent approximately 14 days of the spawn run period (the first mycelial development) in the production room (Figure 1). During this period, the production room was not supplied with fresh air and carbon dioxide (CO_2_) concentration inside the room was kept above 5000 ppm. The compost temperature was regulated between 24 and 26 °C. After this period, in order to lay an equal thickness of cover soil on the composts, the compost surfaces were flattened. The cover soil with a thickness of about 5 cm was laid on the compost surfaces on the 15th day and at this time, the second mycelial development started.

At the end of the second mycelial development period of approximately 6–7 days, when fungal mycelia begin to appear on the surface of the cover soil or after developing in about 3/4 of the soil thickness (mycelial dressing), both to aerate the compacted cover soil and to ensure the homogeneous mycelium development in the cover soil, raking was carried out. The raking process was carried out to the depth where the cover soil merged with the compost by mixing. After the raking process, the same climatic conditions mentioned above were applied in the room in order to reunite them with each other to ensure mycelia were broken during the raking process. When it was observed that the broken mycelium on the cover soil surface converged and the development of the mycelia continued, the temperature of the production room was gradually lowered from 21–23 °C to 16–17 °C within 3–4 days. With this application, transition of mushrooms from the vegetative development to generative development period is possible. It means that the period has been started for the formation of the stem and cap of the mushroom, which we call the “*carpophore*”. As the room temperature began to decrease, fresh air was gradually introduced into the production room (Figure 1).

About 35 days after the compost entered the production room, the first flash started and was completed in an average of 4–5 days. After the completion of the first flash, the harvest of the mushrooms in the production room was completed and the production room was discharged at the end of the second flash period, which lasted for 4–5 days after a one-week flash break (Figure 2).

The yield and quality criteria of the mushrooms harvested during the flash periods, as well as the shelf life of the mushrooms stored at different temperatures after the harvest (1st flash), and the quality criteria of this period were examined.

### 2.2. Storage and Shelf Life Conditions

After harvest, mushrooms obtained from different PGPR applications were placed in plastic plates and stored at 1 ± 1 °C, 85–90% relative humidity conditions for 15 days. During the cold storage period and at 5-day intervals, samples were transferred from the storage room to the shelf life room having 7 ± 1 °C temperature, 60–70% relative humidity conditions for 2 days.

### 2.3. Parameters

#### 2.3.1. Mushroom Weight and Cap Number, Yield

The average mushroom weight was determined by dividing the yield from each plot by the number of mushrooms.

The number of mushrooms was determined by counting the plots of all treatments.

The total yield was calculated as grams (g) by weighing the mushrooms harvested from each plot with a digital scale sensitive to 0.05 g.

The yield percentage was calculated by comparing obtained yield to 100 kg compost.

#### 2.3.2. Cap and Stem Characteristics

Cap and stem diameters were determined by a digital caliper in mm from the widest part of the cap and stem after removing the cap from the stem.

After removing the cap from the stem, the cap was placed on a flat plane and the length between bottom and the top side was measured by a digital caliper in mm.

#### 2.3.3. Color

Fifteen mushrooms per replication were used for this parameter. Color measurements were carried out at the middle point of the cap by a chromameter (CR-400, Minolta Co., Tokyo, Japan) in the CIE *L*, a*, b** color system [25]. By using these values, chroma (*C**), hue (h°) [25], color difference (∆*E_L*,a*,b_*_*_) [26], browning index (BI) [27], yellowness index (YI) [28] were calculated according to the equations given below.
Chroma (*C**) = (*a**^2^ + *b**^2^)^1/2^(1)
Hue (*h°*) = arctan (*b**/*a**)(2)
∆*E_L*_*_,*a**,*b**_ = [(*L_t_** − *L_i_**) + (*a_t_** − *a_i_**) + (*b_t_** − *b_i_**)]^1/2^(3)
∆*E* indicates the degree of overall color change. *L_i_**, *a_i_** and *b_i_** refer to the values at the beginning of storage period *L_t_**, *a_t_** and *b_t_** refer to readings during storage and shelf life periods.
(4)$$\begin{array}{c} \mathrm{Browning}\;\mathrm{index}\;(\mathrm{BI})=100\times [X-0.31)/0.17]\\ \mathrm{where}\\ X=\left[a^{*}+\left(1.75\times L^{*}\right)\right]/\left[\left(5.645\times L^{*}\right)+a^{*}-\left(3.012\times b^{*}\right)\right] \end{array}$$
Yellowness index (YI) = (142.86 × *b**)/*L*.*(5)

#### 2.3.4. Cap Firmness

After removing a thin layer from the upper part of the cap, firmness was determined by a fruit texture meter (Fruit Texture Analyzer, GS-15, GÜSS Manufacturing Ltd., North Africa) using a probe having a 7.9 mm diameter, and the results were presented as Newton (N) force.

#### 2.3.5. Water Content

A certain amount of mushroom samples taken from each replication was firstly weighed with a digital scale and then dried in an oven at 65 °C (UM400, Memmert, Germany) until the weight was stabilized, and the amount of water was determined as % by weighing the samples again.

#### 2.3.6. Total Phenolics and Antioxidant Capacity

Extraction from mushrooms in order to determine the total phenol content and antioxidant activity was described by Thaipong et al. [29]. The content of total phenolic substances was determined by the Folin–Ciocalteu method [30]. In this method, gallic acid was used as a standard and the total amount of phenolic substance in the mushroom was given as mg gallic acid equivalent (GAE) per 100 g fresh weight (FW). Antioxidant capacity was measured according to the FRAP method and the results were presented as μmol Trolox equivalents (TE) per g FW [31].

#### 2.3.7. Weight Loss

Weight loss was explained on the basis of the initial weights. For this reason, at the beginning of the storage period 15 mushrooms were separated for each replication. The same samples were weighed by a precision scale and the weights of the samples were compared to initial weights. The data were presented as percentages.

#### 2.3.8. Experimental Design and Statistical Evaluations

The study was carried out in four replications according to a randomized plot design, and each replication (plot) consisted of 2 plastic boxes, approximately 25 × 35 cm in size, with a volume of 19 L and containing 7 × 2 = 14 kg compost. For storage and shelf life periods, a randomized experimental design with four replications was applied and each replication included 15 mushrooms.

The data obtained from the experiment were subjected to analysis of variance (ANOVA) using the statistical package program MINITAB (trial version). In addition to each storage period, the differences between mean values of shelf life were controlled by Tukey’s test using the MSTAT-C program (Michigan State University, East Lansing) at *p* ≤ 0.05 error level. Arcsin transformations were applied to percentage data before statistical analysis. Moreover, in order to realize the relationship among the investigated quality parameters, Pearson correlation tests were applied to data.

## 3. Results and Discussion

### 3.1. Harvest Characteristics

After harvest, mushroom weight, total yield and yield percentage, cap diameter and height and stem diameter were determined before the storage period in order to investigate the effect of PGPRs (Table 1). The effects of different PGPR applications on the average mushroom weight of white button mushrooms were non-significant (*p* > 0.05) and the mushroom weights varied between 16.46 g and 17.73 g.

Different PGPR applications significantly affected the number of caps of mushrooms (*p* ≤ 0.05) (Table 1). The highest cap number (203.78) was in the AL inoculated group and the lowest value was 142.00 in the TT inoculated group. The total yield was the highest in the AL applied group (3474.00 g per 14 kg compost) (Table 1). However, controls, BM and FA were not significantly different from the AL applied group. The TT inoculated group had with the lowest yield of 2402.00 g per 14 kg compost. This application seems to prevent increases in yield, as it was not different from the control group. Similarly, yield percentages were the highest in the AL inoculated group (24.81%) followed by BM (21.45%), controls (18.60%) and FA (18.44%), respectively. The lowest value was calculated in the TT inoculated group (17.16%).

Inoculation of compost by nitrogen bacteria (*Azospillum lipoferum*, AL) increased the mushroom yield by approximately 33.3% compared to the controls. This very significant increase in yield was achieved by the increase in the number of mushrooms taken from the unit area (28.3%). The differences in yield observed among applications are due to the differences in the number of caps. The significant increase in the number of caps by *Azospillum lipoferum* inoculation can be explained by the fact that nitrogen in the compost can be become available. It is thought that this effect of nitrogen bacteria was observed in fruit and vegetable production as well as in growth hormones of white button mushroom, germination, root development, nutrient uptake, productivity and resistance to abiotic stress conditions [7,8]. Since mushroom harvest is carried out when caps reach a certain size, it is an expected development that mushroom weights will show similarity. Moreover, similar weight values are also an indication that the harvest is carried out at the optimum time.

Different PGPR applications did not significantly affect cap diameter, cap height and stem diameter. This could result from harvesting of the mushrooms that reached a certain size because the optimum harvest criterion for mushrooms is size and a selective harvesting method is preferred. For this reason, mushrooms that do not reach the desired size are not harvested, so the product is of homogeneous size.

### 3.2. Changes in Quality Parameters during Cold Storage and Shelf Life

#### 3.2.1. Color

*L** value refers lightness of a color. While the effect of different applications on the cap *L** value of the mushrooms was significant (*p* ≤ 0.05) after 10 and 15 days of storage, in addition to the shelf life, this effect was insignificant in the previous storage periods (Table 2). After 10 days of the storage period plus a shelf life period of two days, the control group showed the lowest *L**, but after a storage period of 15 days plus 2 days of shelf life, the lowest value was determined in the BM inoculated group. On this date, the highest *L** values were observed in AL, FA and TT inoculated samples. Based on the average values, the lowest *L** level was in controls and was the highest in the AL inoculated group. However, all *L** values measured in the PGPR inoculated groups were in the same statistical group. As the storage period continued, significant color changes occurred due to aging. Mushroom color is one of the most important indicators in consumer preference. An *L** value lower than 80 reduces the marketability of mushrooms [32,33] and a value less than 69 is considered unmarketable for product [34]. In the current study, AL and FA inoculations led to relatively little change in *L** values during the storage period. It can be said that these treatments can help to maintain quality during shelf life periods after a certain duration of cold storage. This situation can also be interpreted as these applications slowing down aging in white button mushroom.

During shelf life periods, *C** and hue values in different PGPR inoculated samples significantly changed depending on the individual effects of the factors (applications and storage period) (*p* ≤ 0.05) (Table 2). During the storage period *C** values tended to increase while hue values showed a linear decrease. Within the applications, the control group showed the highest average *C** value (19.55) followed by FA (19.22) and AL (18.02) treated groups. The lowest *C** value (7.33) was observed in BM treated samples. Higher *C** values refer to aging in mushroom samples and it seems that firstly BM, secondly TT and then AL treatments inhibit aging in white button mushroom. On the other hand, the lowest hue value in TT applied samples showed that TT inoculation accelerated aging in white button mushroom. However, all other PGPR treatments inhibited the decrease in hue values and they indirectly inhibited aging of the samples. The white color turns to a yellow-brown color in white button mushrooms during the process of aging [35].

#### 3.2.2. Color Changes

The color changes observed in mushrooms during storage varied depending on the individual effects of storage period and PGPR applications (Table 3).

As the storage period continued, the ∆*E_L*,a*,b*_* value as well as the browning index and the yellowness index increased. The highest values in all three parameters were determined in the samples kept under shelf conditions for 2 days in addition to 15 days of cold storage. On the other hand, a stable effect of PGPR applications on the change in these parameters was not observed. When the mean values of the treatments were examined, the highest ∆*E_L*,a*,b*_* value (6.03) was measured in the BM treated group, and the lowest (3.64) in the FA treated group. This is an indication that FA application has a very limited effect on color change during the whole storage process.

The highest browning index value (27.16) was found in the control group samples. This group was followed by TT applied samples (Table 3). On the other hand, statistically lower browning index values were determined in AL, BM and FA applied groups. Higher browning index values indicate that aging is more advanced in white button mushrooms (*p* ≤ 0.05).

Browning of mushrooms is affected by many factors, including intrinsic and extrinsic factors. The important factors affecting enzymatic browning of button mushrooms are mechanical damage, low relative humidity and toxins of pathogens. These factors can lead to stimulating the breakdown of the intracellular membrane due to contact between substrates and enzyme. On the other hand, environmental conditions such as higher oxygen pressure and temperature cause increased browning [35]. During the marketing period of white button mushrooms, prevention of color changes will offer a big advantage.

In this case, it can be said that the applications of AL, BM and FA relatively inhibit aging in mushrooms under shelf conditions after a certain period of cold storage (Table 3). Similarly, observation of higher yellowness index values with the progression of the storage period indicates that these values increase in aging mushrooms as in browning index. However, the lowest yellowness index values were determined in the BM treated group followed by AL and TT treated groups (*p* ≤ 0.05) (Table 3). FA application resulted in negative effects with respect to browning index and yellowness index values in white button mushrooms. There are some edible coatings to inhibit color changes during the storage period [32], but this type of treatment will need additional labor and result in an increase in expenses. In this respect, PGPR applications during the production stage can offer a solution to inhibit significant color changes during the marketing or storage period of white button mushrooms.

#### 3.2.3. Firmness, Water Content, Weight Loss

Cap hardness or firmness refers to the mechanical resistance of white button mushrooms against external factors. It is an important quality parameter in terms of postharvest shelf life. The effect of different PGPRs inoculated into compost, the growing medium, on the cap firmness of the mushrooms was significant (*p* ≤ 0.05) in all shelf life periods after a certain duration of cold storage (Table 4).

However, differences among PGPR applications were not significant after a shelf life period of 2 days following 15 days of cold storage. Generally, control and AL, nitrogen bacteria, applied samples showed the lowest firmness values till the end of the 10 + 2 days of the shelf life period and additionally these applications had the lowest firmness values with respect to mean values in all applications, as well. On the other hand, firmness of mushrooms inoculated by BM, phosphorus bacteria, was similar to those treated with FA, potassium bacteria. It is thought that the low cap firmness of the mushrooms in the controls and AL inoculated samples is related to the cellular status of the cap. The size of the cells and the weak strength of the intercellular connections in horticultural products cause low firmness [36]. The fact that AL inoculation is effective in increasing the cell size in the cap could be responsible for low firmness values [24]. During the storage period, there was a steady decrease in the cap firmness of mushrooms in all treatments. This decrease is compatible with the aging of the mushroom cap. Due to the process of aging in the cap, some organic substances such as pectin, cellulose and hemicellulose that give rigidity to the cap become soluble and the intercellular connections weaken, and eventually loss in firmness occurs [37].

The effects of different bacterial treatments on the water content of the fungi were similar in all storage periods (*p* ≤ 0.05) (Table 4). The amount of water changed between 94.02% and 95.28% at the beginning of storage (0 + 2 days) and varied between 91.36% and 92.50% at the end of storage (15 + 2 days). The water content values detected in mushrooms inoculated with different PGPRs changed depending on the interactive effects of the storage time and PGPR applications. Generally, the lowest water content was observed in the BM inoculated group on 0th + 2, 5th + 2 and 15th + 2 days of storage life. Moreover, considering the average values, the BM inoculation resulted in the lowest water content. On the other hand, controls and AL inoculated groups showed similar and higher water content values. During the storage period, water content in mushrooms decreased regularly and, in all groups, the lowest values were determined on the 15th + 2 day. This situation could have resulted from an increase in weight loss on the same day.

The interactive effects of storage time and PGPR applications were important in the change in water loss values in the current study (*p* ≤ 005) (Table 4). In all PGPR applied groups and controls, weight loss values increased during storage time. This is quite normal because all samples were stored in open conditions and also in the open during shelf life periods. The highest weight loss rates generally occurred on the 10th + 2 day and 15th + 2 day. The differences observed among PGPR applications were particularly significant from the 5th + 2 day. While the highest weight losses were observed in the groups inoculated with FA and TT, respectively, the lowest values were observed in the control and AL groups on 5th + 2 and 10th + 2 days. On the other hand, AL application led to the highest weight loss on the 15th + 2 day. Although TT had the lowest weight loss rate, it was not significantly different from the controls. The average values showed that controls and BM applied samples had lower weight loss, while other PGPR applications helped to increase this parameter. For this parameter, within the PGPR applications, it seems that BM is more successful at maintaining weight loss in white button mushroom.

#### 3.2.4. Total Phenolics and Antioxidant Capacity

It was determined that the effect of different PGPR applications on the total phenol content of the mushrooms was significant (*p* ≤ 0.05) based on average values for applications and storage periods (Table 5). The average amount of phenol was the highest in mushrooms treated with phosphorus bacteria, the BM group, after the shelf life period, and the lowest in the control group. The total phenol content of the fungi treated with TT and FA was similar to those treated with BM. This can be explained as BM, FA and TT inoculations improving the nutrition of the mushrooms during the growth and development period. A decrease in the phenol content was observed as the storage period progressed. Enzymes associated with aging could be effective in this decrease. Indeed, studies have shown that water loss in mushrooms is directly related to polyphenoloxidase activity [38,39].

The individual effect of the factors in this research, storage period and PGPR applications, is significant for antioxidant capacity of the mushrooms (*p* ≤ 0.05) (Table 5). For the average values, longer storage period caused a decrease in antioxidant capacity. The highest antioxidant capacity was measured in FA inoculated samples as 11.21 μmol TE g FW^−1^ but data of BM (10.86 μmol TE g FW^−1^) and TT (1.85 μmol TE g FW^−1^) groups were in the same statistical group. It means that these PGPR inoculations stimulated an increase in antioxidant capacity. On the other hand, control and AL, nitrogen bacteria, applied groups were similar with respect to antioxidant capacity. It has been thought that the positive effect of PGPR inoculations on antioxidant capacity in white button mushroom could be related to better nutrition during growth and development. Slow aging during storage in well-fed mushrooms is effective in the preservation of phytochemicals [40]. As a matter of fact, it was observed that color and hardness changes were slower in mushrooms treated with *p*, K and S bacteria during storage. In general, it was observed that the effect of bacteria on the total phenol content and antioxidant activity of fungi was positive.

#### 3.2.5. Relationships among Investigated Parameters

In this study, all quality parameters investigated during storage and shelf life periods were related to each other (*p* ≤ 0.05) (Table 6). There is one exception, that weight loss and antioxidant capacity were not correlated with each other. This could have resulted from the method used for determination of antioxidant capacity. Browning index is a very important quality criterion in white button mushroom storage. It mostly affects consumer preference. In the current study, browning index data were well correlated with *L**, *C**, *hue*, ∆*E_L*,a*,b*_*, yellowness index data. It means that apart from browning index, other color parameters and color change markers can be used to follow the physiological stage or aging in white button mushrooms. Moreover, these parameters can also be used to guess some other parameters such as firmness, water content, total phenolic content and antioxidant capacity in white button mushrooms.

## 4. Conclusions

In white button mushroom cultivation, inoculation of the growing medium, compost, with *Azospillum lipoferum* (AL) nitrogen bacteria, increased the number of mushrooms per unit area, resulting in a significant increase in yield. While a low yield increase is achieved with the different production techniques, after the current study, it is thought that the increase in mushroom yield, especially in commercial enterprises, will increase the profitability by reducing the production cost, thanks to the nitrogen bacterial inoculation. During cold storage and shelf life periods, phosphorus and potassium bacteria, *Bacillus megaterium* (BM) and *Frateuria aurantia* (FA), respectively, showed limited loss in the cap firmness. Especially in the last period of storage, phosphorus and potassium bacteria slowed down the weight loss and resulted in limited color changes in the mushroom caps. It was determined that *Bacillus megaterium* (BM, phosphorus bacteria), *Frateuria aurantia* (FA, potassium bacteria) and *Thiobacillus thiooxidans* (TT, sulfur bacteria) positively affected total phenolic content and antioxidant capacity in white button mushrooms. The effect of different PGPR inoculations into the compost on the physical properties (weight, diameter, height, length) of the mushroom cap was limited. The results showed that *Azospillum lipoferum* helped to increase in yield, and *Bacillus megaterium* and *Frateuria aurantia* in particular improved the maintenance of quality during a shelf life period of 2 days after a cold storage period of 15 days. Moreover, the practical advantages of these treatments can be summarized as follows: it is possible to apply PGPRs at the beginning of the production stages and, in this way, it can be possible to offer to the consumer a chemical-free product. PGPR treatments can affect the mushroom yield as well as quality, and further postharvest treatments in order to maintain quality during the storage period or to extend storage life may not be necessary. In this case, PGPR applications can help to reduce production costs.

## Figures and Tables

**Figure 1 jof-08-01016-f001:**
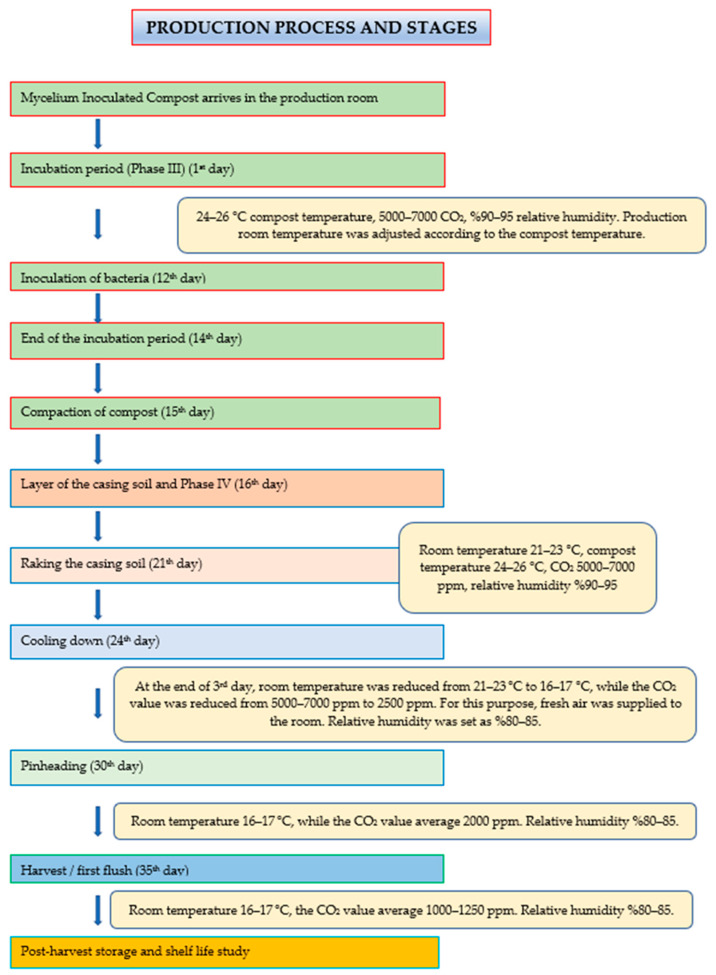
Production stages of *Agaricus bisporus* L.

**Figure 2 jof-08-01016-f002:**
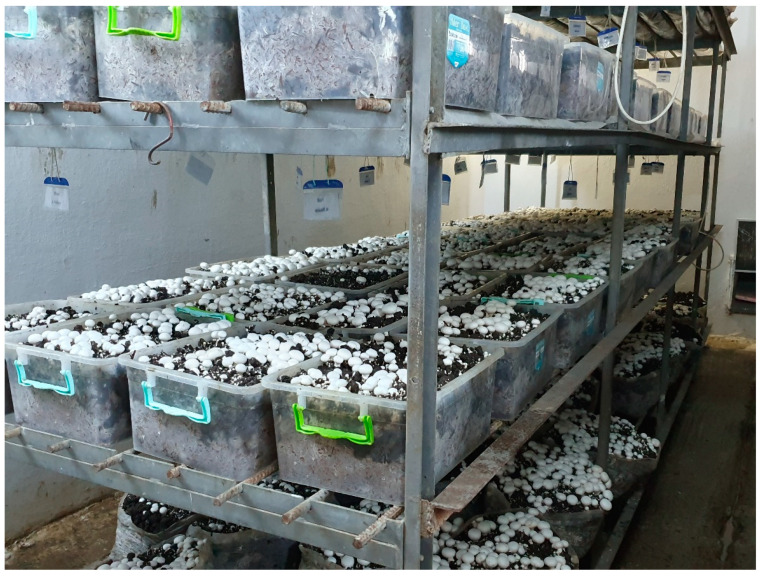
Experimental production room.

**Table 1 jof-08-01016-t001:** Mushroom weight, total yield and yield percentage, cap diameter and height and stem diameter were determined before storage period.

PGPR Applications ^1^	Mushroom Weight (g)	No. Caps	Total Yield (g 14 kg^−1^ Compost)	Yield (%)	Cap Diameter (mm)	Cap Height (mm)	Stem Diameter (mm)
Control	16.46 ± 0.65 ^ns 2^	158.90 ± 24.70 ab ^3^	2604.00 ± 391.00 ab	18.60 ± 2.79 ab	48.60 ± 2.10 ^ns^	23.53 ± 1.75 ^ns^	18.68 ± 0.82 ^ns^
AL	17.07 ± 0.38	203.78 ± 8.38 a	3474.00 ± 118.00 a	24.81 ± 0.84 a	49.66 ± 1.19	24.30 ± 0.60	18.20 ± 0.80
BM	16.94 ± 1.35	177.10 ± 10.10 ab	3003.00 ± 308.00 ab	21.45 ± 2.20 ab	51.90 ± 6.60	25.54 ± 1.69	20.06 ± 2.04
FA	17.73 ± 0.07	145.60 ± 3.79 b	2581.50 ± 61.500 ab	18.44 ± 0.44 ab	51.08 ± 1.62	25.10 ± 0.85	18.50 ± 0.30
TT	17.10 ± 1.37	142.00 ± 24.80 b	2402.00 ± 419.00 b	17.16 ± 2.99 b	49.43 ± 2.13	23.65 ± 1.37	18.03 ± 1.87
*Variation*	*sources*						
Applications (*p*)	0.911	0.007	0.030	0.030	0.107	0.350	0.118

^1^ Data are presented as mean ± standard error of mean (SEM), *n* = 4; PGPR applications; AL, *Azospillum lipoferum*; BM, *Bacillus megaterium*; FA, *Frateuria aurantia*; TT, *Thiobacillus thiooxidans*; ^2^ ns, non-significant at *p* ≤ 0.05 error level; ^3^ letters show differences among PGPR applications for each parameter at *p ≤* 0.05 error level.

**Table 2 jof-08-01016-t002:** The effect of different PGPR applications on color parameters in white button mushroom during shelf life periods.

PGPR Applications ^1^	Shelf Life Periods (Days)				Average
	0 + 2 ^1^	5 + 2	10 + 2	15 + 2	
	*L**				
Control	90.06 ± 0.78 A,a ^2^	86.46 ± 0.33 A,b	83.85 ± 0.35 B,c	83.29 ± 0.44 B,c	85.91 ± 0.83 B ^3^
AL	90.99 ± 0.93 A,a	87.13 ± 0.46 A,b	85.90 ± 0.29 AB,b	86.23 ± 0.27 A,b	87.56 ± 0.65 A
BM	91.98 ± 0.49 A,a	87.57 ± 0.91 A,b	86.50 ± 1.37 A,b	83.28 ± 0.57 C,c	87.33 ± 1.01 A
FA	89.67 ± 0.34 A,a	87.17 ± 0.96 A,b	85.75 ± 0.43 AB,b	85.46 ± 0.32 AB, b	87.01 ± 0.55 AB
TT	90.59 ± 0.40 A,a	87.06 ± 0.52 A,b	87.25 ± 1.31 A,b	84.65 ± 0.29 AB,c	87.39 ± 0.65 A
Average	90.65 ± 0.32 a ^4^	87.08 ± 0.27 b	85.85 ± 0.40 c	84.58 ± 0.35 d	
*Variation Sources*					*p*
*Shelf Life Periods*					0.000
*PGPR Applications*					0.003
*Shelf Life Periods X PGPR Applications*					0.026
	*C**				
Control	15.65 ± 0.46 ^ns 5^	19.35 ± 0.30 ^ns^	21.83 ± 0.17 ^ns^	21.34 ± 0.21 ^ns^	19.55 ± 0.75 A
AL	14.31 ± 0.82	19.37 ± 0.75	19.48 ± 0.54	18.88 ± 0.28	18.02 ± 0.70 ABC
BM	13.64 ± 0.66	17.61 ± 0.92	18.29 ± 1.92	19.75 ± 0.20	17.33 ± 0.83 C
FA	16.86 ± 0.32	18.60 ± 1.45	20.85 ± 0.55	20.55 ± 0.36	19.22 ± 0.59 AB
TT	14.93 ± 0.70	17.91 ± 0.95	18.67 ± 0.62	20.04 ± 0.16	17.89 ± 0.63 BC
Average	15.08 ± 0.38 c	18.57 ± 0.41 b	19.83 ± 0.051 ab	20.11 ± 0.24 a	
*Variation Sources*					*p*
*Shelf Life Periods*					0.000
*PGPR Applications*					0.001
*Shelf Life Periods X Applications*					0.474
	*h°*				
Control	86.34 ± 0.55 ^ns^	83.55 ± 0.16 ^ns^	82.96 ± 0.06 ^ns^	83.06 ± 0.41 ^ns^	83.98 ± 0.44 A ^3^
AL	86.74 ± 0.74	84.32 ± 0.56	84.98 ± 0.61	84.64 ± 0.31	85.18 ± 0.37 A
BM	87.55 ± 1.94	84.76 ± 0.60	85.36 ± 0.98	82.65 ± 0.03	85.08 ± 0.71 A
FA	85.52 ± 0.17	85.25 ± 1.20	84.34 ± 0.53	84.22 ± 0.15	84.83 ± 0.33 A
TT	86.97 ± 1.23	85.96 ± 0.77	86.30 ± 0.16	84.21 ± 0.32	85.86 ± 0.44 B
Average	86.63 ± 0.46 a	84.77 ± 0.36 b	84.79 ± 0.37 b	83.76 ± 0.23 b	
*Variation Sources*					*p*
*Shelf Life Periods*					0.000
*PGPR Applications*					0.018
*Shelf Life Periods X Applications*					0.375

^1^ Data were presented as mean ± standard error of mean (SEM), *n* = 4; PGPR applications; AL, *Azospillum lipoferum*; BM, *Bacillus megaterium*; FA, *Frateuria aurantia*; TT, *Thiobacillus thiooxidans*; days in cold storage at 1 °C + days on shelf at 20 °C; ^2^ uppercase letters show differences among PGPR applications in each shelf life period, lowercase letters show differences among shelf life periods in each PGPR application; ^3^ uppercase letters show differences among average data of PGPR applications; ^4^ lowercase letters show differences among average data of shelf life periods at *p* ≤ 0.05 error level; ^5^ ns, non-significant at *p* ≤ 0.05 error level.

**Table 3 jof-08-01016-t003:** The effect of different PGPR applications on color changes in white button mushroom during shelf life periods.

PGPR Applications ^1^	Shelf Life Periods (Days)				Average
	0 + 2 ^1^	5 + 2	10 + 2	15 + 2	
	∆*E_L*,a*,b*_*				
Control	0.00 ± 0.00 ^ns 2^	5.24 ± 0.49 ^ns^	8.83 ± 0.70 ^ns^	8.93 ± 0.90 ^ns^	5.75 ± 1.13 AB ^3^
AL	0.00 ± 0.00	6.42 ± 0.39	7.30 ± 1.08	6.66 ± 1.28	5.09 ± 0.96 AB
BM	0.00 ± 0.00	6.06 ± 2.08	7.33 ± 2.57	10.74 ± 1.41	6.03 ± 1.40 A
FA	0.00 ± 0.00	3.28 ± 1.91	5.61 ± 0.28	5.65 ± 0.40	3.64 ± 0.81 B
TT	0.00 ± 0.00	5.04 ± 0.27	5.17 ± 1.05	7.91 ± 0.82	4.53 ± 0.91 AB
Average	0.0 ± 0.0 c ^4^	5.21 ± 0.57 b	6.85 ± 0.63 ab	7.98 ± 0.61 a	
*Variation Sources*					*p*
*Shelf Life Periods*					0.000
*PGPR Applications*					0.023
*Shelf Life Periods X PGPR Applications*					0.380
	Browning Index				
Control	19.47 ± 0.95 ^ns^	26.49 ± 0.57 ^ns^	31.63 ± 0.42 ^ns^	31.03 ± 0.48 ^ns^	27.16 ± 1.49 A
AL	17.42 ± 1.39	26.13 ± 1.42	26.54 ± 1.01	25.58 ± 0.28	23.91 ± 1.23 B
BM	16.17 ± 1.30	23.29 ± 1.71	24.67 ± 3.67	28.51 ± 0.56	23.16 ± 1.63 B
FA	21.39 ± 0.51	24.83 ± 2.81	28.90 ± 1.17	28.55 ± 0.45	25.92 ± 1.14 B
TT	18.25 ± 1.19	23.56 ± 1.52	24.52 ± 0.94	28.05 ± 0.30	23.60 ± 1.15 AB
Average	18.54 ± 0.64 c	24.86 ± 0.75 b	27.25 ± 1.00 ab	28.34 ± 0.49 a	
*Variation Sources*					*p*
*Shelf Life Periods*					0.000
*PGPR Applications*					0.001
*Shelf Life Periods X PGPR Applications*					0.301
	Yellowness Index				
Control	24.79 ± 0.94 ^ns^	31.78 ± 0.56 ^ns^	36.92 ± 0.41 ^ns^	36.34 ± 0.47 ^ns^	32.46 ± 1.49 A
AL	22.47 ± 1.49	31.62 ± 1.36	32.28 ± 0.92	31.15 ± 0.40	29.38 ± 1.30 BC
BM	21.15 ± 1.11	28.64 ± 1.74	30.21 ± 3.63	33.61 ± 0.57	28.40 ± 1.64 C
FA	26.78 ± 0.62	30.41 ± 2.66	34.57 ± 1.08	34.18 ± 0.52	31.49 ± 1.15 AB
TT	23.52 ± 1.12	29.31 ± 1.47	30.51 ± 1.05	33.65 ± 0.40	29.25 ± 1.20 BC
Average	23.74 ± 0.66 c	30.35 ± 0.72 b	32.90 ± 0.97 a	33.80 ± 0.48 a	
*Variation Sources*					*p*
*Shelf Life Periods*					0.000
*PGPR Applications*					0.001
*Shelf Life Periods X PGPR Applications*					0.355

^1^ Data are presented as mean ± standard error of mean (SEM), *n* = 4; PGPR applications; AL, *Azospillum lipoferum*; BM, *Bacillus megaterium*; FA, *Frateuria aurantia*; TT, *Thiobacillus thiooxidans*; days in cold storage at 1 °C + days on shelf at 20 °C; ^2^ ns, non-significant at *p* ≤ 0.05 error level; ^3^ uppercase letters show differences among average data of PGPR applications; ^4^ lowercase letters show differences among average data of shelf life periods at *p* ≤ 0.05 error level.

**Table 4 jof-08-01016-t004:** The effect of different PGPR applications on firmness, water content and weight loss in white button mushroom during shelf life periods.

PGPR Applications ^1^	Shelf Life Periods (Days)				Average
	0 + 2 ^1^	5 + 2	10 + 2	15 + 2	
	Firmness (N)				
Control	26.95 ± 0.81 BC,a ^2^	28.01 ± 0.42 C,a	22.23 ± 0.63 B,b	22.69 ± 0.51 A,b	24.97 ± 0.81 BC ^3^
AL	26.62 ± 0.02 C,a	26.69 ± 1.02 C,a	21.38 ± 1.05 B,b	22.57 ± 0.48 A,b	24.32 ± 0.79 C
BM	33.68 ± 0.70 A,a	33.74 ± 0.45 A,a	26.22 ± 0.15 A,b	23.27 ± 0.80 A,b	29.23 ± 1.41 A
FA	33.20 ± 0.57 A,a	32.55 ± 1.34 AB,a	28.05 ± 1.25 A,b	24.54 ± 0.36 A,c	29.59 ± 1.14 A
TT	30.43 ± 1.77 AB,a	29.51 ± 1.44 BC,a	24.55 ± 0.51 AB,b	22.69 ± 1.46 A,b	26.80 ± 1.14 B
Average	30.13 ± 0.88 a ^4^	30.10 ± 0.81 a	24.49 ± 0.73 b	23.15 ± 0.37 b	
*Variation Sources*					*p*
*Shelf Life Periods*					0.000
*PGPR Applications*					0.000
*Shelf Life Periods X PGPR Applications*					0.037
	Water Content (%)				
Control	94.77 ± 0.02 A,a ^2^	94.43 ± 0.17 A,a	92.20 ± 0.27 A,b	92.03 ± 0.07 AB,b	93.36 ± 0.38 AB ^3^
AL	95.28 ± 0.07 A,a	93.34 ± 0.43 B,b	92.57 ± 0.08 A,c	92.50 ± 0.11 A,c	93.42 ± 0.35 A
BM	94.02 ± 0.13 B,a	94.63 ± 0.01 A,a	92.09 ± 0.15 A,b	91.36 ± 0.06 B,c	93.03 ± 0.41 B
FA	94.74 ± 0.03 AB,a	94.28 ± 0.18 A,a	92.45 ± 0.07 A,b	91.73 ± 0.32 B,c	93.30 ± 0.38 AB
TT	94.57 ± 0.15AB,a	94.86 ± 0.25 A,a	92.01 ± 0.05 A,c	91.91 ± 0.15 C,b	93.34 ± 0.42 AB
Average	94.67 ± 0.11 a ^4^	94.30 ± 0.17 b	92.27 ± 0.08 c	91.90 ± 0.11 d	
*Variation Sources*					*p*
*Shelf Life Periods*					0.000
*PGPR Applications*					0.029
*Shelf Life Periods X PGPR Applications*					0.000
	Weight Loss (%)				
Control	0.79 ± 0.10 A,c ^2^	2.21 ± 0.10 C,b	2.76 ± 0.33 C,ab	3.44 ± 0.11 BC,a	2.31 ± 0.37 C ^3^
AL	0.77 ± 0.07 A,c	2.52 ± 0.11 C,b	3.13 ± 0.09 C,b	4.98 ± 0.22 A,a	2.85 ± 0.57 AB
BM	0.86 ± 0.08 A,c	2.69 ± 0.09 BC,b	3.48 ± 0.31 BC,a	3.55 ± 0.23 BC,a	2.65 ± 0.42 BC
FA	0.86 ± 0.01 A,c	3.46 ± 0.18 A,b	4.21 ± 0.44 AB,a	4.05 ± 0.17 B,ab	3.15 ± 0.52 A
TT	0.87 ± 0.09 A,c	3.28 ± 0.13 AB,b	4.26 ± 0.43 A,a	3.14 ± 0.19 C,b	2.90 ± 0.48 AB
Average	0.84 ± 0.83 c ^4^	2.84 ± 2.84 b	3.57 ± 3.57 a	3.83 ± 3.83 a	
*Variation Sources*					*p*
*Shelf Life Periods*					0.000
*PGPR Applications*					0.000
*Shelf Life Periods X PGPR Applications*					0.000

^1^ Data are presented as mean ± standard error of mean (SEM), *n* = 4; PGPR applications; AL, *Azospillum lipoferum*; BM, *Bacillus megaterium*; FA, *Frateuria aurantia*; TT, *Thiobacillus thiooxidans*; days in cold storage at 1 °C + days on shelf at 20 °C; ^2^ uppercase letters show differences among PGPR applications in each shelf life period, lowercase letters show differences among shelf life periods in each PGPR application; ^3^ uppercase letters show differences among average data of PGPR applications; ^4^ lowercase letters show differences among average data of shelf life periods at *p* ≤ 0.05 error level.

**Table 5 jof-08-01016-t005:** The effect of different PGPR applications on total phenolic content and antioxidant capacity in white button mushroom during shelf life periods.

PGPR Applications ^1^	Shelf Life Periods (Days)				Average
	0 + 2 ^1^	5 + 2	10 + 2	15 + 2	
	Total Phenolic Content (mg GAE 100 g FW^−1^)				
Control	66.43 ± 3.27 ^ns 2^	59.83 ± 2.96 ^ns^	56.13 ± 1.55 ^ns^	48.82 ± 1.81 ^ns^	57.80 ± 2.20 B ^3^
AL	69.08 ± 2.37	65.31 ± 4.42	59.51 ± 3.93	53.25 ± 1.97	61.79 ± 2.30 AB
BM	71.15 ± 4.33	70.43 ± 2.90	67.22 ± 5.19	56.08 ± 1.86	66.22 ± 2.43 A
FA	69.51 ± 2.12	65.00 ± 1.66	59.48 ± 2.35	52.68 ± 0.21	61.67 ± 2.04 AB
TT	69.13 ± 2.86	70.52 ± 2.00	63.93 ± 5.35	56.48 ± 0.38	65.02 ± 2.15A
Average	69.06 ± 1.24 a ^4^	66.22 ± 1.54 ab	61.25 ± 1.82 b	53.46 ± 0.92 c	
*Variation Sources*					*p*
*Shelf Life Periods*					0.000
*PGPR Applications*					0.003
*Shelf Life Periods X PGPR Applications*					0.989
	Antioxidant Capacity (μmol TE g FW^−1^)				
Control	11.22 ± 0.47 ^ns^	9.09 ± 0.04 ^ns^	8.24 ± 0.37 ^ns^	8.81 ± 1.40 ^ns^	9.34 ± 0.47 C ^3^
AL	11.44 ± 1.25	9.53 ± 0.70	8.96 ± 0.13	8.77 ± 0.59	9.67 ± 0.46 BC
BM	11.42 ± 0.28	10.82 ± 0.80	12.00 ± 0.11	9.21 ± 0.27	10.86 ± 0.37 AB
FA	11.11 ± 0.33	11.78 ± 1.28	11.44 ± 0.52	10.51 ± 0.90	11.21 ± 0.39 A
TT	10.91 ± 0.59	12.27 ± 0.33	11.81 ± 1.08	8.44 ± 0.67	10.85 ± 0.54 AB
Average	11.22 ± 0.26 a ^4^	10.69 ± 0.44 a	10.49 ± 0.47 a	9.15 ± 0.38 b	
*Variation Sources*					*p*
*Shelf Life Periods*					0.000
*PGPR Applications*					0.029
*Shelf Life Periods X PGPR Applications*					0.000

^1^ Data were presented as mean ± standard error of mean (SEM), *n* = 4; PGPR applications; AL, *Azospillum lipoferum*; BM, *Bacillus megaterium*; FA, *Frateuria aurantia*; TT, *Thiobacillus thiooxidans*; days in cold storage at 1 °C + days on shelf at 20 °C; ^2^ ns, non-significant at *p* ≤ 0.05 error level; ^3^ uppercase letters show differences among average data of PGPR applications; ^4^ lowercase letters show differences among average data of shelf life periods at *p* ≤ 0.05 error level.

**Table 6 jof-08-01016-t006:** Relationships among the quality parameters followed during storage and shelf life periods.

Parameters	*L**	*C**	Hue	∆*E_L*,a*,b*_*	BI ^1^	YI	F	WC	WL	TPC
*C**	*** ^2^									
*Hue*	***	***								
∆*E_L*,a*,b*_*	***	***	***							
BI	***	***	***	***						
YI	***	***	***	***	***					
F	***	***	***	***	***	***				
WC	***	***	***	***	***	***	***			
WL	***	***	***	***	***	***	***	***		
TPC	***	***	***	***	***	***	***	***	**	
AOC	***	**	**	**	**	**	***	**	ns	**

^1^ BI, browning index; YI, yellowness index; F, firmness (N); WC, water content (%); WL, weight loss (%); TPC, total phenolic content (mg GAE 100 g FW^−1^); AOC, antioxidant capacity (μmol TE g FW^−1^); ^2^ Pearson correlation quotients, r^2^ = −1
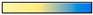
+1; ***, **, * refer to *p* ≤ 0.001, *p* ≤ 0.01, *p* ≤ 0.05, respectively; ns, non-significant at *p* ≤ 0.05 error level.

## Data Availability

All data relevant to the study are included in the article.

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
