# Peer review of "The Effect of Plant Growth Promoting Rhizobacteria (PGPRs) on Yield and Some Quality Parameters during Shelf Life in White Button Mushroom (Agaricus bisporus L.)"

_jof, 2022, doi:10.3390/jof8101016_

Round 1

Reviewer 1 Report

This study has research significance for the cultivation of Agaricus bisporus. There are two suggestions: 1. It is recommended that the author make a schematic diagram according to the methods, so that readers can understand it better; 2. It is recommended that the author add pictures of the fruiting bodies in the results, such as 2.3.2, 2.3.3 et al. The results are more convincing and clear.

Reviewer 2 Report

Comments and suggestions are highlighted in yellow in ms (attached file)

Author Response

Dear Reviewer,

I would like to thank you for your kind efforts. 

Best regards,
